# Galectin-3 deficiency drives lupus-like disease by promoting spontaneous germinal centers formation via IFN-γ

Cristian Gabriel Beccaria[1], María Carolina Amezcua Vesely[1], Facundo Fiocca Vernengo[1], Ricardo Carlos Gehrau[1], María Cecilia Ramello[1], Jimena Tosello Boari[1], Melisa Gorosito Serrán[1], Juan Mucci [2], Eliane Piaggio[3,4], Oscar Campetella[2], Eva Virginia Acosta Rodríguez [1], Carolina Lucía Montes[1] & Adriana Gruppi [1]

Germinal centers (GC) are important sites for high-affinity and long-lived antibody induction. Tight regulation of GC responses is critical for maintaining self-tolerance. Here, we show that Galectin-3 (Gal-3) is involved in GC development. Compared with WT mice, Gal-3 KO mice have more GC B cells and T follicular helper cells, increased percentages of antibody-secreting cells and higher concentrations of immunoglobulins and IFN-γ in serum, and develop a lupus-like disease. IFN-γ blockade in Gal-3 KO mice reduces spontaneous GC formation, class-switch recombination, autoantibody production and renal pathology, demonstrating that IFN-γ overproduction sustains autoimmunity. The results from chimeric mice show that intrinsic Gal-3 signaling in B cells controls spontaneous GC formation. Taken together, our data provide evidence that Gal-3 acts directly on B cells to regulate GC responses via IFN-γ and implicate the potential of Gal-3 as a therapeutic target in autoimmunity.

[1] Centro de Investigaciones en Bioquímica Clínica e Inmunología (CIBICI)-CONICET, Facultad de Ciencias Químicas, Universidad Nacional de Córdoba, X5000HUA Córdoba, Argentina. [2] Instituto de Investigaciones Biotecnológicas (IIB-INTECH), Universidad Nacional de San Martín (UNSAM) - CONICET, B1650HMP San Martín, Buenos Aires, Argentina. [3] INSERM U932, 75005 Paris, France. [4] Institut Curie, Section Recherche, 75005 Paris, France. These authors contributed equally: María Carolina Amezcua Vesely, Facundo Fiocca Vernengo. Correspondence and requests for materials should be addressed to A.G. (email: agruppi@fcq.unc.edu.ar)

B cell function provides a key line of defence against infections through the production of high affinity antibodies, which are critical for clearing pathogens and preventing reinfections. These antibodies are the main product of long-lived antibody-secreting cells (ASCs) or plasma cells (PCs), which are generated with the help of a distinct subset of CD4$^+$ T cells called T follicular helper (Tfh) cells, within a specialised microenvironment known as the germinal center (GC)[1–3]. The GC contains highly proliferative B cells that undergo Ig somatic hypermutation and isotype switching, in which B cells are also selected based on their antigen affinity. Indeed, GCs are an important B cell–tolerance checkpoint in the periphery. IL-4 and IL-21 are the most important cytokines for assisting B cells during the differentiation process to GC B cells, and mice that are deficient for both IL-4 and IL-21 receptors ($Il4^{-/-}Il21r^{-/-}$ mice) were found to have severe defects in antibody production and GC formation[4].

Although GC activity is fundamental for health by inducing protective immunity, overreactivity of the GC pathway causes autoimmunity. Dysregulated GC B cell and Tfh responses make decisive contributions to the generation of class-switched autoantibodies and to the development of lupus in several mouse models[5–8]. Associations between autoimmunity and overactive GCs have also been suggested in human systemic lupus erythematosus (SLE), Sjogren's syndrome, rheumatoid arthritis, and autoimmune thyroid disease[9]. Vinuesa and collaborators[10] have demonstrated that the reduction of GC cell numbers caused by the specific deletion of one allele of $Bcl6$ ameliorates the autoimmune pathology observed in $Roquin^{san/san}$ (sanroque) mice, which develop GCs in the absence of foreign antigen. A reduction of spontaneous GC formation via the blockade of T cell signaling in different autoimmune experimental models also reduces disease[6,7,11]. Thus, a fine equilibrium between GC formation and number is required for tolerance.

GC formation and B cell differentiation to PCs is strictly regulated by several transcription factors[12]. Bcl-6 and Blimp-1 are transcriptional repressors with the ability to block each other's expression. Bcl-6 is expressed by GC B cells and is required for this phase of B cell development. Bcl-6-deficient mice lack GC B cells and affinity maturation. Constitutive expression of Bcl-6 in B cells in vivo results in a large GC. In addition, three studies have shown that Bcl-6 is a master regulator of Tfh differentiation[13–15]. PC differentiation, in contrast, critically depends on Blimp-1 expression and the absence of Bcl-6. Blimp-1 is highly expressed in PCs and controls many genes that are important for PC differentiation. Furthermore, this transcription factor also inhibits genes involved in cellular proliferation, such as $Myc$ and $Bcl6$, thereby allowing the terminal differentiation of PC[16].

Although much progress has been made in understanding the factors and rules that govern the generation of GCs and the formation of long-lived ASCs during infections and autoimmunity, the identification of new molecules is crucial to achieve a better understanding of this complex process. This is a pivotal issue not only for the design of vaccines against pathogens and malignant cells but also for therapeutic intervention in autoimmunity.

We and others have shown that the downregulation or deficit of Galectin-3 (Gal-3) leads to an increase in the number of ASCs during the course of *Trypanosoma cruzi* and *Schistosoma mansoni* infection[17,18] and promotes peritoneal B1 cell differentiation into PCs[19]. Moreover, B cells with reduced expression of endogenous Gal-3 fail to down-regulate Blimp-1 after IL-4 stimulation[17]. Hoyer and collaborators have demonstrated that Gal-3 is present in naive and memory B cells but almost absent in differentiated B cells, such as CD10$^-$/IgD$^-$ GC B cells, and PCs[20].

Taken together, these findings support the hypothesis that Gal-3 may influence GC and ASC generation.

Gal-3 belongs to the large growing family of highly conserved β-galactoside binding lectins known as galectins. Gal-3 is the only chimera-type galectin composed of an unusual non-lectin proline and glycine-rich domain coupled to a carbohydrate-recognition domain[21]. Gal-3 has a broad distribution among different types of cells and tissues. It can be localised intracellularly, in the nucleus and cytoplasm, or extracellularly, secreted via the non-classical pathway[22]. Relying on its expression level, the type of cell and the specific immune condition, Gal-3 can be either a positive or a negative regulator of the immune response[23–25]. Considering that Gal-3 is expressed in B cells[17,20] and T cells and that the downregulation of Gal-3 improves antigen-specific antibody production[17], we explored whether Gal-3 could influence the GC reaction and B cell differentiation into ASCs, processes that involve a delicate crosstalk between B and T cell populations.

Our study reveals previously unrecognised functions of Gal-3 in the regulation of GC responses. The absence of Gal-3 in mice drives an excess of IFN-γ, which leads to aberrant GC formation and autoantibody production. This study reveals Gal-3 as a key factor in the development of IFN-γ-mediated lupus-like disease.

## Results

**Gal-3 has a critical function in GC formation.** To investigate the Gal-3 expression pattern in differentiated B cells, purified splenic B cells were stimulated with LPS or anti-CD40 plus IL-4 to induce differentiation into ASCs. Figure 1a shows that Gal-3 protein was detectable in splenic B cells and its expression decreased to undetectable and very low levels in LPS-stimulated and anti-CD40 plus IL-4-activated B cells, respectively. To define the kinetics and pattern of Gal-3 downregulation upon B cell differentiation induced by LPS, splenic B cells were cultured alone or with LPS for different times. While the Gal-3 protein level rose in B cells cultured for 48 and 72 h without stimuli, it became completely undetectable after 48 h of LPS stimulation (Fig. 1a).

As LPS is an inducer of PC differentiation[26], the above-described results suggest that a decrease in Gal-3 expression can be associated with antibody secretion. Thus, we next compared Gal-3 expression between peritoneal B1 and splenic B2 cells with differential abilities to spontaneously secrete Igs. We observed no differences in Gal-3 expression among these two B cell subsets (Supplementary Fig. 1a). Additionally, Gal-3 was also detectable in T cells, and this expression was substantially lower than in B cells (Supplementary Fig. 1a). In agreement with previous reports[27], specific fluorescence staining for Gal-3 was detected in both the nucleus and cytoplasm of non-stimulated resting B cells (Supplementary Fig. 1b, upper panels), and a remarkable loss of fluorescence intensity was observed when B cells were stimulated with LPS (Supplementary Fig. 1b, lower panel). These findings indicate that the differentiation of B cells to ASCs is not associated with a sub-cellular redistribution of Gal-3 protein but rather a reduction of protein synthesis. Indeed, stimulation of B cells with LPS induced a down-regulation of the transcript encoding Gal-3 (Supplementary Fig. 1c). Gal-3 mRNA levels were significantly down-regulated after 24 and 48 h of LPS stimulation with respect to the transcript level at 12 h ($p < 0.001$ by one-way ANOVA). In agreement with the protein expression level (Fig. 1b), B cells cultured in the absence of stimulation had similar Gal-3 mRNA levels at 12 and 24 h, but a significant upregulation ($p < 0.01$ by one-way ANOVA) was detected after 48 h of culture with medium (Supplementary Fig. 1c).

To address the role of Gal-3 in B cell immunobiology in vivo, we took advantage of Gal-3 KO mice. We first compared the concentration of serum antibodies and the frequency of splenic

and bone marrow ASCs in Gal-3 KO and WT mice. We found that, without purposeful immunization or infection, Gal-3 KO mice displayed significantly ($p < 0.001$ by unpaired Student's $t$-test) higher concentrations of IgM, IgG2c, and IgG3 in serum (Fig. 1c), as well as higher frequencies of IgM and IgG ASCs in spleen and bone marrow (Fig. 1d), in comparison to WT mice. Considering that production of antibodies under homoeostatic conditions relies on different B cells subsets and can be produced by ASCs derived or not from the GC, we examined the frequency of different mature B cells (follicular, FO, and marginal zone, MZ, B cells), GC B cells and PCs in spleen from Gal-3 KO and WT mice. We found no significant differences ($p > 0.05$ by unpaired

Student's $t$-test) in the absolute number of FO and MZ B cells between the groups of mice (Fig. 1e), but Gal-3 KO mice had a significantly greater number of GC (B220$^+$ FAS$^+$ GL7$^+$ CD3$^-$ CD11b$^-$ CD11c$^-$) and PC (B220$^{int}$ CD138$^+$ CD3$^-$) B cells (Fig. 1f–h) compared with their WT counterparts ($p < 0.001$ and $p < 0.01$ by unpaired Student's $t$-test, respectively). The phenotype of GC B cells was confirmed by low levels of IgD and CD38 (Supplementary Fig. 2a–c) and high levels of CXCR4 and CD40 expression, and a high frequency of Ki-67$^+$ cells within B220$^+$ GL7$^+$ FAS$^+$ B cells, in comparison to the phenotype exhibited by non-GC B cells (B220$^+$ GL7$^-$ FAS$^-$) (Supplementary Fig. 2d–f, respectively). Immunofluorescence analysis of spleen sections

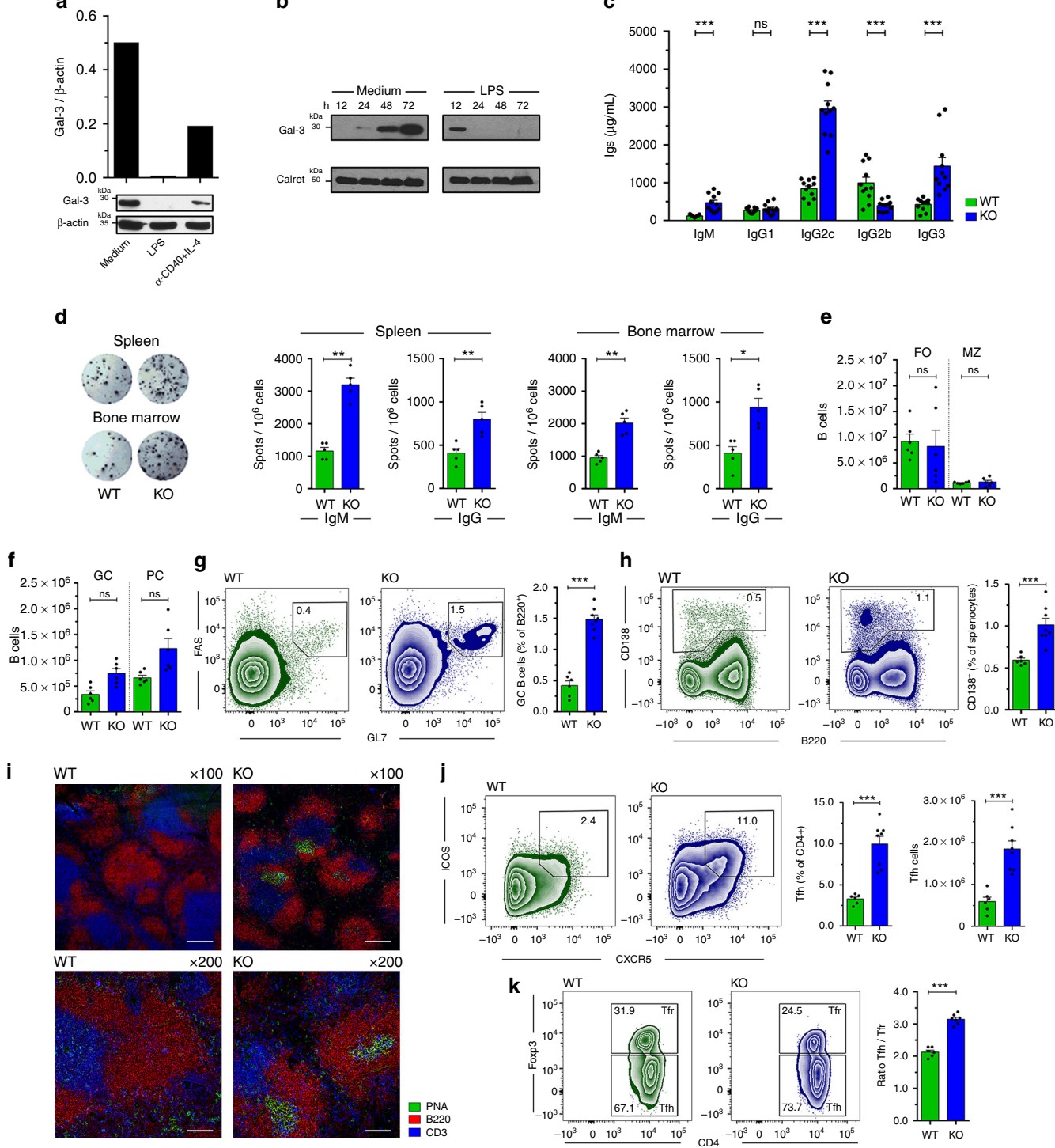

**Fig. 1** Downregulation of Gal-3 during PC differentiation and increased concentration of Igs and spontaneous GC formation in Gal-3 KO mice. **a** Splenic B cells were cultured with medium, LPS, or anti-CD40 plus IL-4 for 72 h, and Gal-3 expression was evaluated by western blot analysis. The Gal-3 signal was normalised to β-Actin ($n = 5$ for each condition). **b** Splenic B cells were cultured medium or LPS from 12 to 72 h. The Gal-3 signal was normalised to calreticulin ($n = 3$ for each condition). **c** Quantification by ELISA of IgM, IgG1, IgG2c, IgG2b, and IgG3 in sera from 8-week-old WT and Gal-3 KO ($n = 11$ for both strains) mice. **d** Representative images (left) from the ELISPOT assay of IgG-ASCs in the spleen and BM and statistical analysis (right) of IgM and IgG ASCs in the spleen and BM from 8-week-old WT and Gal-3 KO mice ($n = 5$ for both strains). **e** Numbers of splenic MZ B cells and FO B cells determined by flow cytometry in WT and Gal-3 KO mice ($n = 6$ for both strains). **f** Numbers of splenic GC B cells and PCs determined by flow cytometry in WT and Gal-3 KO mice ($n = 6$ for both strains). **g, h** Representative plots and statistical analysis of splenic GC B cells (**g**) and PCs cells (**h**) in WT ($n = 6$) and Gal-3 KO ($n = 8$) mice. **i** Immunofluorescence of spleen sections (7 μm) from WT and Gal-3 KO mice, indicated as KO, stained with PNA (green), anti-B220 (red), and anti-CD3 (blue). Magnification: ×100 (top) and ×200 (bottom) ($n = 6$ for both strains). Scale bar: 200 μm (upper panels), 100 μm (lower panels). **j** Representative plots and statistical analysis of splenic Tfh cells in WT and Gal-3 KO mice ($n = 6$ for both strains). **k** Representative plots and statistical analysis of the ratio between splenic Tfh and Tfr cells in WT ($n = 6$) and Gal-3 KO ($n = 8$) mice. Data are the mean ± s.e.m. *$p < 0.05$, **$p < 0.005$, ***$p < 0.001$. ns, not significant (unpaired Student's $t$-test). **a–k** Data are representative of three independent experiments. WT mice are indicated in green and Gal-3 KO mice in blue

revealed larger spontaneous PNA+⁺ GCs within the follicles in Gal-3 KO mice, whereas WT mice showed a complete absence of GCs (Fig. 1i).

When we analysed the presence of Tfh cells, identified as CD4⁺ CXCR5⁺ ICOS⁺ Foxp3⁻ T cells, in Gal-3 KO mice, we observed that, in clear contrast to the few Tfh cells observed in WT mice, the spleen of Gal-3 KO mice exhibited an exacerbated frequency of Tfh cells (Fig. 1j). We verified the Tfh cell identity based on the high PD-1 and Bcl-6 expression (Supplementary Fig. 3a–d). We also examined Tfr cells, a subset of CD4⁺ Foxp3⁺ Treg cells within the GC that coopt a CXCR5⁺ ICOS⁺ phenotype and limit the magnitude and output of the GC response. Among CD4⁺ CXCR5⁺ ICOS⁺ T cells, the frequency of Tfr cells was reduced in Gal-3 KO mice in comparison to WT (Fig. 1k). However, the total Treg (CD4⁺ Foxp3⁺ CD25⁺) frequency and number were similar in both groups of mice (Supplementary Fig. 4a), and there were no differences in expression of the regulatory molecules CTLA-4 or CD39 on Treg cells between WT and Gal-3 KO mice (Supplementary Fig. 4b, c, respectively). Of note, the Tfh/Tfr

ratio, which is an important factor in humoral immunity because it dictates the magnitude of antibody responses[28], was increased in Gal-3 KO mice. Taken together, these results indicate that the absence of Gal-3 drives spontaneous GC formation and, consequently, Igs production.

**Gal-3-deficient B cells express GC-associated molecules.** To investigate whether Gal-3 affects the transcriptional programme of B cells, we performed gene expression profiling by microarray using purified B cells from spleens of WT and Gal-3 KO mice (Fig. 2a). Heat maps highlighted the most differentially expressed transcript of genes that are relevant for GC biology (Fig. 2a). Remarkably, the levels of transcripts encoding Igs and molecules associated with the GC B cell phenotype and function, such as *Gcet2*[29] and *H2B*[30]; the indicator of proliferation *Mki67* and the critical component of antibody class-switching *Aicda* were upregulated in B cells from Gal-3 KO mice. In accordance with spontaneous GC formation, the array analysis showed an upregulation of two genes encoding miRs associated with the GC

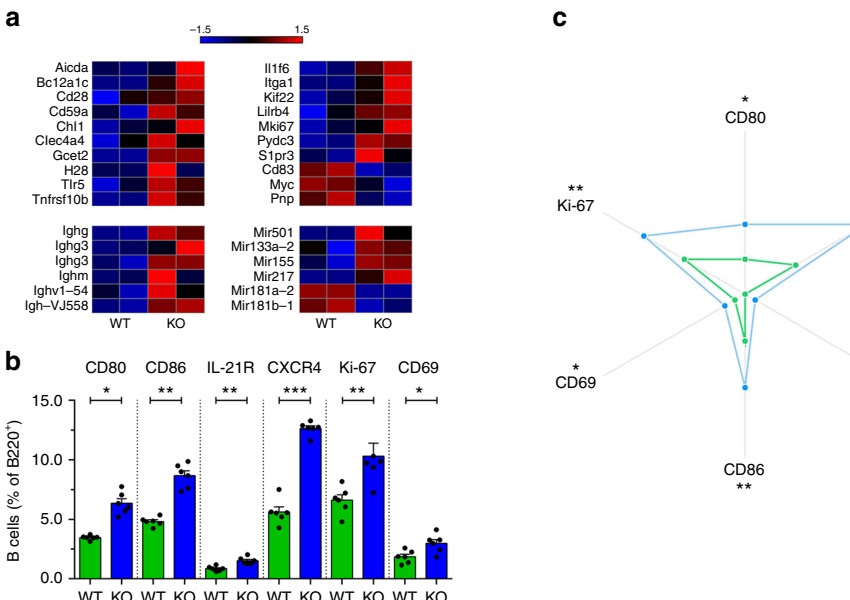

**Fig. 2** B cells from Gal-3 KO mice had increased expression of genes related to GC formation and costimulatory molecules. **a** Heat maps of the expression of selected genes related to the GC reaction determined in B cells from WT and Gal-3 KO mice ($n = 2$ for both strains). **b** Statistical analysis of the frequency of CD80⁺, CD86⁺, IL-21R⁺, CXCR4⁺, Ki-67⁺, and CD69⁺ B220⁺ B cells from the spleen of 8-week-old WT and Gal-3 KO mice ($n = 6$ for both strains). **c** Radar plots summarising the B cell phenotype observed in the spleen of Gal-3 KO and WT mice ($n = 6$ for both strains). Data are the mean ± s.e.m. *$p < 0.05$, **$p < 0.005$ (unpaired Student's $t$-test). **a** Data are representative of two independent experiments. **b, c** Data are representative of three independent experiments

reaction: *miR-155*[31,32] and *miR-217*[33]. The absence of Gal-3 in B cells resulted in a downregulation of *miR-181b-1*, the decrease of which is also associated with the GC response and lupus[34]. The gene and surface molecule expression profile of Gal-3 KO-derived B cells highlighted that the absence of Gal-3 favors an activation phenotype that is prone to differentiate into GC B cells. To gain further insights into the overall impact of the absence of Gal-3

signaling on B cells, we determined the expression of markers involved in the GC reaction. We observed that IL-21R, a key cytokine receptor involved in the regulation of the Bcl-6 and GC response[35], was expressed at a higher frequency on B cells from Gal-3 KO in comparison to WT (Fig. 2b, c). We also found a higher percentage of CD80-expressing and CD86-expressing B cells in Gal-3 KO mice compared with their WT counterparts

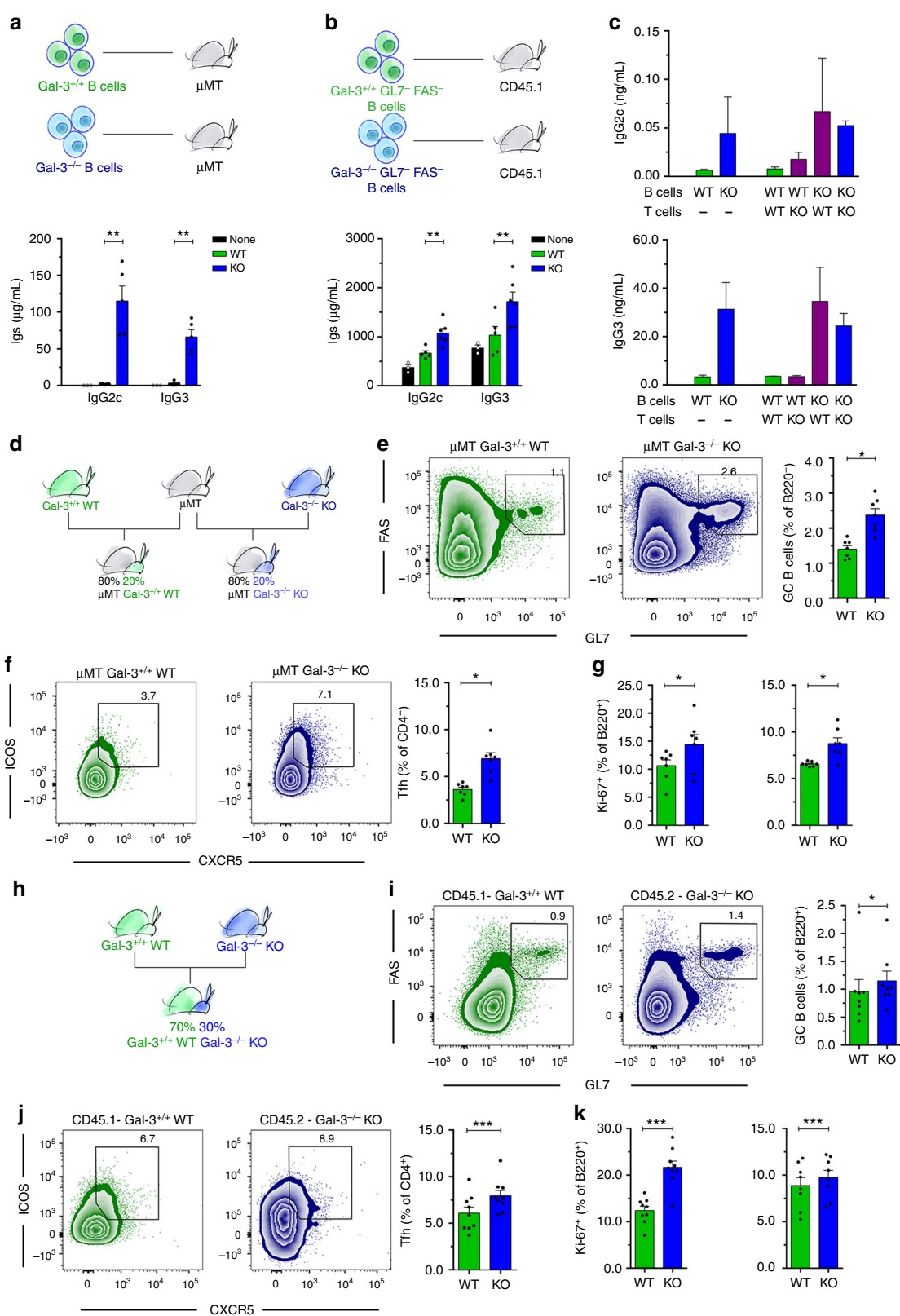

**Fig. 3** B cell-intrinsic Gal-3 deficiency favors ASC differentiation in vivo and in vitro and triggers spontaneous GC formation. **a** Schematic of the adoptive transfer experiment showing that B cells from Gal-3$^{+/+}$ WT and Gal-3 KO mice were injected iv into μMT mice. Bar graph showing the seric IgG2c and IgG3 concentration of recipient μMT mice at 2 weeks after cell transfer (μMT+Gal-3$^{+/+}$ B cells $n = 5$; μMT+Gal-3$^{-/-}$ B cells $n = 5$). **b** Schematic of the adoptive transfer experiment showing that non-GC B cells from Gal-3$^{+/+}$ WT and Gal-3 KO mice were injected iv into CD45.1 WT mice. Bar graph showing the seric IgG2c and IgG3 concentrations of recipient WT mice at 2 weeks after cell transfer (CD45.1+Gal-3$^{+/+}$ B cells $n = 5$; CD45.1+Gal-3$^{-/-}$ B cells $n = 6$). **c** IgG2c and IgG3 concentrations in the supernatants of B cells co-cultured with T cells (ratio 1:1) from the spleen of WT or Gal-3 KO mice and stimulated for 6 days with anti-CD3 and anti-IgM. The combinations of B and T cells are indicated ($n = 4$ for each condition). **d** Schematic of mixed BM chimeric mice containing a 80:20 ratio of μMT:Gal-3$^{+/+}$ WT or μMT:Gal-3 KO BM cells. **e, f** Representative plots and statistical analysis of the frequency of splenic GC B cells (**e**) and Tfh cells (**f**) from mixed BM chimeric mice shown in Fig. 3d ($n = 7$ for each group). **g** Frequency of splenic Ki-67$^+$ B220$^+$ B cells (left) and Ki-67$^+$ CD4$^+$ T cells (right) from mixed BM chimeric mice shown in Fig. 3d ($n = 7$ for each group). **h** Schematic of mixed BM chimeric mice containing a 70:30 ratio of CD45.1 Gal-3$^{+/+}$ WT:CD45.2 Gal-3 KO BM cells. **i, j** Representative plots and statistical analysis of the frequency of splenic GC B cells (**i**) and Tfh cells (**j**) from mixed BM chimeric mice shown in Fig. 3h ($n = 9$). **k** Frequency of splenic Ki-67$^+$ B220$^+$ B cells (left) and CD4$^+$ T cells (right) from mixed BM chimeric mice shown in Fig. 3h ($n = 9$). Data are the mean ± s.e.m. *$p < 0.05$, **$p < 0.005$, ***$p < 0.001$ (unpaired (**a–h**) and paired (**i–k**) Student's $t$-test). Data are representative of three (**a–c**) or two (**d–k**) independent experiments

(Fig. 2b). In addition, we analysed molecules indicative of the activation status of the cells and observed that Gal-3 KO mice had higher percentages of CD69$^+$ and Ki-67$^+$ B cells (Fig. 2b); the last one is a marker widely used to identify cells undergoing active division, in agreement with the presence of spontaneous GC development. Radar plots summarise the B cell phenotype observed in spleen of Gal-3 KO and WT mice (Fig. 2c) and clearly show that B cells from Gal-3 KO mice exhibit a higher frequency of GC formation-related molecules.

**Gal-3 acts directly on B cells to regulate GC responses.** Next, we wanted to address whether Gal-3-deficient B cells have the capability to spontaneously differentiate and secrete Igs in the absence of Ag stimulation and within a Gal-3-sufficient physiological environment. To achieve this goal, we performed two different adoptive transfer experiments. First, total purified splenic B cells from WT or Gal-3 KO mice were transferred to B cell-deficient μMT hosts, and the levels of antibodies were quantified in sera two weeks after reconstitution (Fig. 3a). Sera from mice transferred with Gal-3-deficient B cells presented a higher concentration of IgG2c and IgG3 than sera from recipients of B cells from WT (Fig. 3a). As expected, sera from μMT that did not receive cells had undetectable Igs.

In a second approach, we transferred sorted GL7$^-$ Fas$^-$ B cells (non-GC B cells) from WT or Gal-3 KO mice into WT recipients (Fig. 3b). Sera from WT mice that received non-GC Gal-3-deficient B cells had a higher concentration of IgG2c and IgG3 than unmanipulated WT mice or WT mice transferred with Gal-3-sufficient non-GC B cells (Fig. 3b).

Next, we assessed the impact of endogenous Gal-3 deficiency on the ability of B cells to produce antibodies in vitro. To achieve this goal, we cultured sorted GL7$^-$ Fas$^-$ B cells from WT or Gal-3 KO mice with sorted CD4$^+$ T cells from WT or Gal-3 KO mice in the presence of anti-IgM and anti-CD3 (Fig. 3c). Gal-3 KO B cells produced larger amounts of IgG2c and IgG3 than WT B cells, independently of the presence of CD4$^+$ T cells from Gal-3 KO or WT mice. All the results described above support the concept that B cell-intrinsic Gal-3 deficiency promotes spontaneous antibody secretion and, probably, spontaneous GC formation.

To directly test this idea, we generated mixed bone marrow (BM) chimeras in which the absence of Gal-3 was limited to the B cell compartment by reconstituting lethally irradiated C57BL/6 recipients with 80% μMT BM together with 20% Gal-3 KO BM (μMT plus Gal-3$^{-/-}$ KO, Fig. 3d). The control group (μMT plus Gal-3$^{+/+}$ WT, Fig. 3d) consisted of irradiated mice reconstituted with an 80:20 mix of μMT and Gal-3-sufficient WT BM. Two months after reconstitution, B cell-restricted Gal-3-KO chimeras developed an evident splenic GC reaction, as reflected by FACS

analysis. A significant increase ($p < 0.05$ by unpaired Student's $t$-test) in Fas$^+$ GL7$^+$ GC B cells (Fig. 3e) and CD4$^+$ CXCR5$^+$ ICOS$^+$ Tfh (Fig. 3f) was observed in the recipients of μMT plus Gal-3$^{-/-}$ KO BM in comparison to the recipients of μMT plus Gal-3$^{+/+}$ WT BM. Similar to Gal-3KO mice, B220$^+$ and CD4$^+$ T cells from B cell-restricted Gal-3 KO chimeras presented higher percentages of Ki-67$^+$ cells than their counterparts in the B cell Gal-3-sufficient chimeras (Fig. 3g).

Next, we decided to assess whether the haematopoietic cells from Gal-3 KO mice could influence the response of Gal-3-sufficient B and T cells from WT mice. To achieve this goal, we developed chimeric mice in which lethally irradiated C57BL/6 mice were reconstituted with a 70:30 mix of CD45.1 Gal-3$^{+/+}$ WT and CD45.2 Gal-3$^{-/-}$ KO BM, respectively (Fig. 3h). After two months of BM cell injection, we detected a significantly higher percentage of B cells with the GC phenotype (B220$^+$ Fas$^+$ GL7$^+$ CD4$^-$) and Tfh (CD4$^+$ CXCR5$^+$ ICOS$^+$), as well as Ki-67$^+$ B220$^+$ B cells and CD4$^+$ T cells, among the CD45.2$^+$ Gal-3$^{-/-}$ KO cells than the CD45.1$^+$ WT counterparts (Fig. 3i–k, respectively) ($p < 0.05$, $p < 0.001$, and $p < 0.001$ by paired Student's $t$-test, respectively). This result indicated than Gal-3 KO cells did not influence B and T cell responses in Gal-3-sufficient mice.

**Increased IFN-γ-producing CD4$^+$ T cells in Gal-3 KO mice.** Because Gal-3 KO mice had a particular serum Ig isotype profile, we investigated whether this pattern correlated with a specific type of T cell response defined by cytokine production. Cytokines secreted by Tfh cells and other cells within the GC niche influence both the growth, survival, and differentiation of ASCs as well as the differentiation of Tfh cells in an autocrine manner[35,36]. We observed that, following anti-CD3/CD28 stimulation, purified CD4$^+$ T cells from Gal-3 KO mice produced significantly higher ($p < 0.01$ by unpaired Student's $t$-test) amounts of IFN-γ, IL-21, and IL-4 and similar levels of IL-6 and IL-2 compared with CD4$^+$ T cells from WT (Fig. 4a). Accordingly, the frequency of splenic IFN-γ-producing CD4$^+$ T cells in Gal-3 KO mice was 2.5-fold higher than that observed in WT controls (Fig. 4b), while no difference in the frequency of IFN-γ-producing CD8$^+$ T cells, DC cells, NK cells or NKT cells (Supplementary Fig. 5a–d, respectively) was observed between the experimental groups.

Interestingly, Gal-3 KO mice also exhibited a higher frequency of IFN-γ-producing B220$^+$ cells than WT controls (Fig. 4c). Additionally, Tfh contained the highest frequency of IFN-γ-producing cells and displayed the highest IFN-γ expression compared with the non-Tfh and Tfr T cell subsets (Fig. 4d). To investigate whether the B cell-intrinsic deficiency in Gal-3 expression affected IFN-γ production, we used the chimeric mice described in Fig. 3d, in which only B cells lacked Gal-3 signaling.

Selective Gal-3 deficiency in B cells increased the frequency of IFN-γ-producing CD4⁺ T cells (Fig. 4e), thus indicating that Gal-3 signaling in B cells is necessary to regulate IFN-γ production by CD4⁺ T cells. Next, we decided to evaluate IFN-γ production in those chimeric mice described in Fig. 3h sharing Gal-3-sufficient cells and Gal-3-deficient cells. We observed an increased frequency of IFN-γ-producing CD4⁺ T cells among Gal-3 KO CD45.2⁺ cells in comparison to Gal-3-sufficient WT CD45.1⁺ cells (Fig. 4f). To analyse how IFN-γ influence B cell responses in Gal-3 KO mice, we first compared the expression of IFN-γR on B and T cells from WT and Gal-3 KO mice. We observed that B

cells exhibited higher expression of INF-γR in comparison to T cells and that INF-γR expression was higher on the cells from Gal-3 KO mice with respect to WT. Interestingly, GC B cells (B220⁺ CD4⁻ FAS⁺ GL7⁺) had higher expression of INF-γR than non-GC B cells (B220⁺ CD4⁻ FAS⁻ GL7⁻; Fig. 4g). Considering that IFN-γ plays a role in T-bet induction in B cells and is crucial for IgG2c class switching[37–39], we also examined T-bet expression in lymphocytes from WT and Gal-3 KO mice. Regarding INF-γR, lymphocytes from Gal-3 KO mice expressed higher levels of T-bet than their counterparts from WT mice (Fig. 4h, i). We found that GC B cells expressed the highest levels

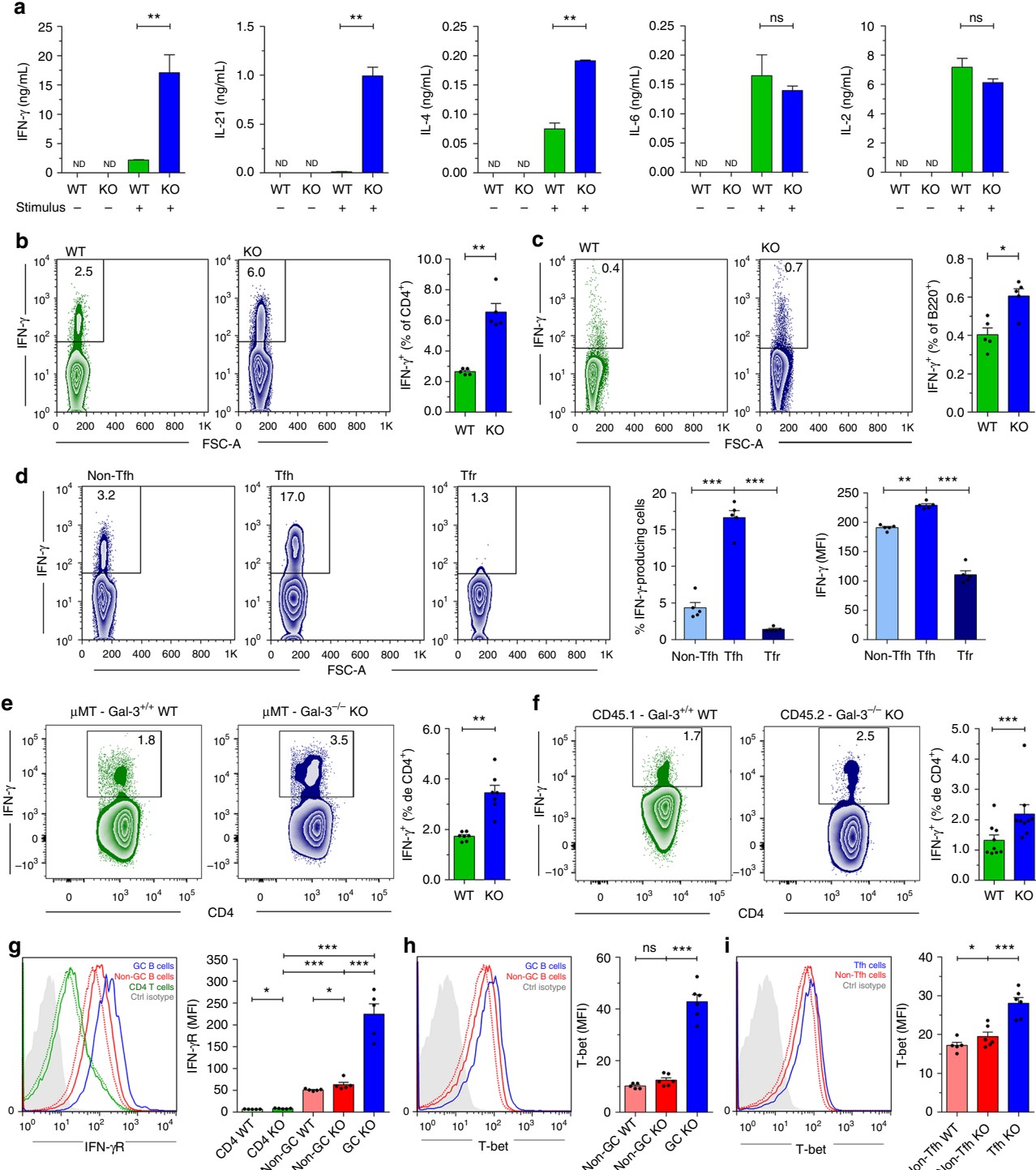

**Fig. 4** Gal-3 KO mice exhibited augmented IFN-γ production. **a** Quantification of IFN-γ, IL-21, IL-4, IL-6, and IL-2 by ELISA in the supernatants of purified CD4$^+$ T cells from the spleen of WT and Gal-3 KO mice cultured with anti-CD3 and anti-CD28 for 72 h ($n = 4$ for each condition). **b** Representative plots and statistical analysis of the frequency of splenic IFN-γ-producing CD4$^+$ T cells of WT and Gal-3 KO mice ($n = 5$ for both strains). **c** Representative plots and statistical analysis of the frequency of splenic IFN-γ-producing B220$^+$ B cells of WT and Gal-3 KO mice ($n = 5$ for both strains). **d** Representative plots and statistical analysis of the frequency of splenic IFN-γ-producing non-Tfh, Tfh, and Tfr cells of Gal-3 KO mice ($n = 5$). **e** Representative plots and statistical analysis of the percentage of splenic IFN-γ-producing CD4$^+$ T cells in the mixed BM chimeras shown in Fig. 3d containing an 80:20 ratio of μMT: Gal-3$^{+/+}$ WT or μMT:Gal-3 KO BM at 10 weeks after transplantation (μMT+Gal-3$^{+/+}$ $n = 7$; μMT+Gal-3$^{-/-}$ B cells $n = 7$). **f** Representative plots and statistical analysis of the percentage of splenic IFN-γ-producing CD4$^+$ T cells in the mixed BM chimeras shown in Fig. 3h containing a 70:30 ratio of CD45.1 Gal-3$^{+/+}$ WT: CD45.2 Gal-3 KO BM at 10 weeks after transplantation ($n = 9$). **g** Representative histograms and statistical analysis of splenic IFN-γR expression on non-GC and GC B cells and on CD4$^+$ T cells of WT (dotted lines) Gal-3 KO mice (solid lines) ($n = 5$ for both strains). **h** Representative histograms and statistical analysis of T-bet expression in splenic non-GC B cells of WT ($n = 5$, dotted lines) and Gal-3 KO mice ($n = 6$, solid lines) and in GCs from Gal-3 KO mice. **i** Representative histograms and statistical analysis of T-bet expression in splenic non-Tfh CD4$^+$ T cells of WT ($n = 5$, dotted lines) and Gal-3 KO mice ($n = 6$, solid lines) and in Tfh from Gal-3 KO mice. Data are the mean ± s.e.m. *$p < 0.05$, **$p < 0.005$, ***$p < 0.001$. ns, not significant (unpaired (**a–e**, **g–i**) and pair (**f**) Student's $t$-test). **a–i** Data are representative of three independent experiments

of T-bet with respect to other populations indicated (Fig. 4h) and that T-bet expression was significantly higher ($p < 0.001$ by unpaired Student's $t$-test) in Tfh than in non-Tfh cells (Fig. 4i).

**Absence of Gal-3 favors autoimmunity.** Spontaneous generation of GC is linked to autoimmunity[5], and IFN-γ has also been associated with spontaneous GC formation and lupus development[40]. Based on our results, it is conceivable that Gal-3 KO mice may develop autoimmune manifestations. Considering that progression of autoimmune diseases is age-related and that secretion of autoantibodies precedes the development of clinical manifestations of lupus[41], we evaluated the presence of antinuclear antibodies (ANA) and tissue damage in 8-month-old Gal-3 KO mice. We observed that Gal-3 mice exhibited higher antibody titres (median 1/640) than WT mice (median 1/40) (Fig. 5a). The most frequent patterns of positivity were as follows: nucleolar, nuclear homogeneous and nuclear fine speckled (Fig. 5c), which are frequently associated with SLE. To confirm the findings of the ANA positivity measured with the HEp-2 assay and to measure isotype- and subclass-specific serum ANA titres, we performed ELISA and immunofluorescence with *Chritidia luciliae* slides for specific nuclear antigens. Gal-3 KO mice had significantly elevated ($p < 0.05$, $p < 0.01$, and $p < 0.001$ by unpaired Student's $t$-test) levels of serum IgM, IgG2c, and IgG3 specific for dsDNA, Sm/RNP and histone, hallmarks of SLE, in comparison to WT mice (Fig. 5b–d). No significant differences ($p > 0.05$ by unpaired Student's $t$-test) were observed in specific IgG1 autoantibodies (Fig. 5d) in serum samples. We also performed a double blinded score study to evaluate kidney damage in Gal-3 KO compared with WT mice. Kidneys from 8-month-old Gal-3 KO mice exhibited an expanded double layer around the glomeruli with thicker tubules, glomerular hypercellularity and hypercellularity in the tubulointerstitium compared with age-matched control mice (Fig. 5e). To characterise the type of infiltrate in the tubulointerstitium, we performed immunofluorescence of kidney slides. We found well-organised lymphoid-like structures wherein B and T cells were intimately interacting with each other (Fig. 5f). Finally, we measured urea and creatinine and 24-h urine proteins, which are important clinical features of renal function. We observed higher concentrations of creatinine and urine proteins in Gal-3 KO compared with WT mice, suggesting a loss of renal function (Fig. 5g, h).

Considering that the emergence of GCs could be age-dependent, we examined the time-course of these structures in both groups of mice. GC B cells were barely detectable in spleen of WT and Gal-3 KO WT mice at weaning (3 weeks old) (Supplementary Fig. 6a), but this cell subset expanded with age and was significantly increased ($p < 0.05$ and $p < 0.01$ by unpaired Student's $t$-test, respectively) in spleen of young adult (8-week-

old) and aged (8-month-old) Gal-3 KO mice. Remarkably, and accompanying the pronounced proliferation of the GC B cell population, Tfh cells were also increased in 8-month-old Gal-3 KO mice in comparison to their counterparts in WT mice (Supplementary Fig. 6a).

Compatible with the marked increase in GC B cells, a significantly higher ($p < 0.05$ by unpaired Student's $t$-test) percentage of B cells from 8-month-old Gal-3 KO mice expressed Bcl-6 (Supplementary Fig. 6b) and exhibited a centroblast phenotype characterised by high expression of CXCR4[42] (Supplementary Fig. 6d). Aged Gal-3 KO mice also had an increased frequency of B cells with the plasmablast/PC phenotype (B220$^+$ CD4$^-$ Blimp-1$^+$) in comparison to WT mice (Supplementary Fig. 6c). Remarkably, B cells from 8-month-old Gal-3 KO presented a higher percentage of CD80$^+$ (Supplementary Fig. 6e) and CD86$^+$ (Supplementary Fig. 6f) cells than their WT counterparts.

In agreement with our previous results, aged Gal-3 KO mice presented abnormally high levels of serum IFN-γ (Fig. 5i). Considering that Gal-3 negatively regulates Th17 polarisation[43] and that Th17 cells may be responsible for aberrant selection of self-reactive GC B cells and autoantibody formation in BXD2 mice[44], we also evaluated IL-17 production in Gal-3 KO mice. No increased levels of IL-17A were detected in sera from aged Gal-3 KO mice (Fig. 5i).

**Blockade of IFN-γ signaling prevents GC reactions and lupus.** We next investigated whether disease can be ameliorated by blocking IFN-γ. To achieve this goal, 4-week-old female Gal-3 KO mice, which have not yet developed GCs, were treated with 500 μg monoclonal antibody (mAb) anti-IFN-γ every 3 days for 3 weeks. At the end of this treatment, anti-IFN-γ-treated mice had reduced numbers of GC B cells and Tfh cells than mice treated with isotype control (Fig. 6a, b, respectively). IFN-γ-blocked mice exhibited a total absence of PNA$^+$ cells inside the follicle (Fig. 6c); as expected, untreated Gal-3 KO mice presented PNA$^+$ GC B cells (Fig. 6c). Interestingly, the follicular B cells of IFN-γ-blocked Gal-3 KO mice expressed IgD, indicating that, in the absence of IFN-γ activity, B cells from Gal-3 KO mice did not experience class-switch recombination (CSR) (Fig. 6d). In addition, IFN-γ blockade in Gal-3 KO mice significantly decreased the frequency of IFN-γ-producing B and T cells (Fig. 6e, f, respectively) ($p < 0.05$ and $p < 0.05$ by unpaired Student's $t$-test, respectively) and the frequency of IFN-γR-expressing non-GC B220$^+$ B cells (Fig. 6g) compared to mice treated with an isotype control. Finally, we determined that anti-IFN-γ-treated Gal-3 mice had undetectable ANA (Fig. 6h) and presented diminished mononuclear cell infiltrates in the kidney (Fig. 6i) in comparison to untreated Gal-3 KO mice. Together, the results demonstrate

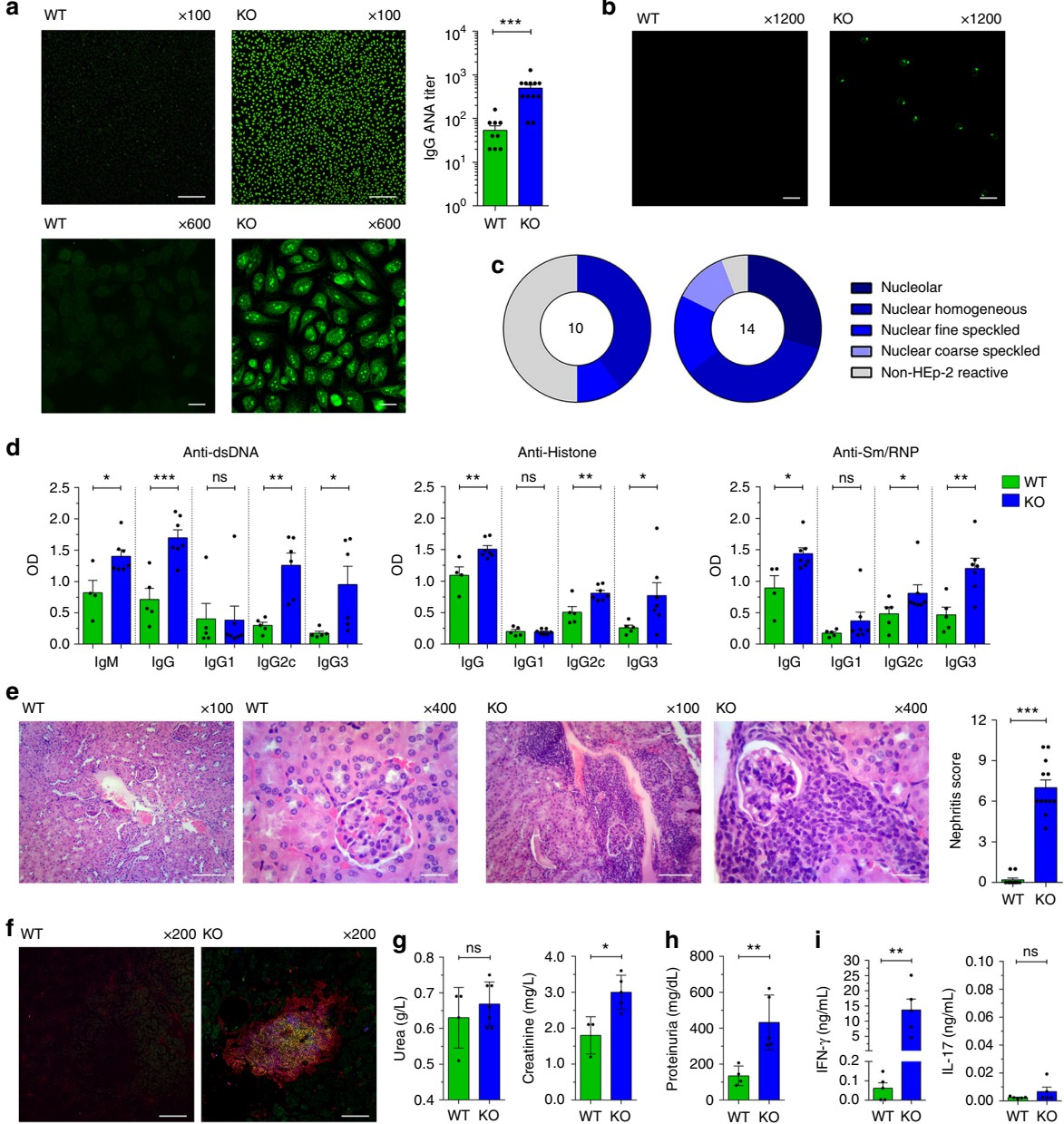

**Fig. 5** Aged Gal-3 KO mice presented signs of autoimmunity. **a** ANA IgG titres determined using Hep-2 slides in sera from 8-month-old WT and Gal-3 KO mice. Magnification: ×100 (top) and ×600 (bottom) (WT $n = 9$; Gal-3 KO $n = 13$). Scale bar, 200 μm (upper panels) and 20 μm (lower panels). **b** dsDNA IgG determined using *Crithidia luciliae* slides. Magnification: ×1200 (WT $n = 9$; Gal-3 KO $n = 13$). Scale bar, 10 μm. **c** Pie charts summarizing the distribution of serum samples exhibiting the different Hep-2 patterns: Nucleolar; nuclear homogeneous; nuclear fine speckled; nuclear coarse speckled; and non-HEp-2 Reactive. The numbers in the centers of the graphs denote the number of samples analysed per group. Animals with ANA titres higher than 1/40 were scored as positive (WT $n = 9$; Gal-3 KO $n = 13$). **d** IgM, IgG1, IgG2c, and IgG3 autoantibodies levels specific for dsDNA, histone and Sm/RNP determined by ELISA (WT $n = 5$; Gal-3 KO $n = 7$). **e** Kidney sections stained with H&E (left) and the nephritis score (right). Magnification: ×100 and ×400 (WT $n = 10$; Gal-3 KO $n = 12$). Scale bar (from left to right), 100, 20, 100, and 20 μm. **f** Immunofluorescence of kidney sections stained with anti-B220 (green), anti-CD45 (red) and anti-CD4 (blue). Magnification: ×200 ($n = 4$ for both strains). Scale bar, 100 μm. **g**, **h** Urea (left) and creatinine (right) concentrations in sera (**g**) and protein levels in 24-h urine samples (**h**) from 8-month-old WT and Gal-3 KO mice (WT $n = 4$; Gal-3 KO $n = 7$). **i** IFN-γ and IL-17 concentrations in sera from 8-month-old WT and Gal-3 KO mice ($n = 5$ for both strains). Data are the mean ± s.e.m. *$p < 0.05$, **$p < 0.005$, ***$p < 0.001$. ns not significant (unpaired Student's $t$-test). All experiments were performed in 8-month-old WT (green) and Gal-3 KO (blue) mice. **a**, **c**, **e** Data are representative of four independent experiments. **b**, **d** Data are representative of two independent experiments. **f–i** Data are representative of three independent experiments

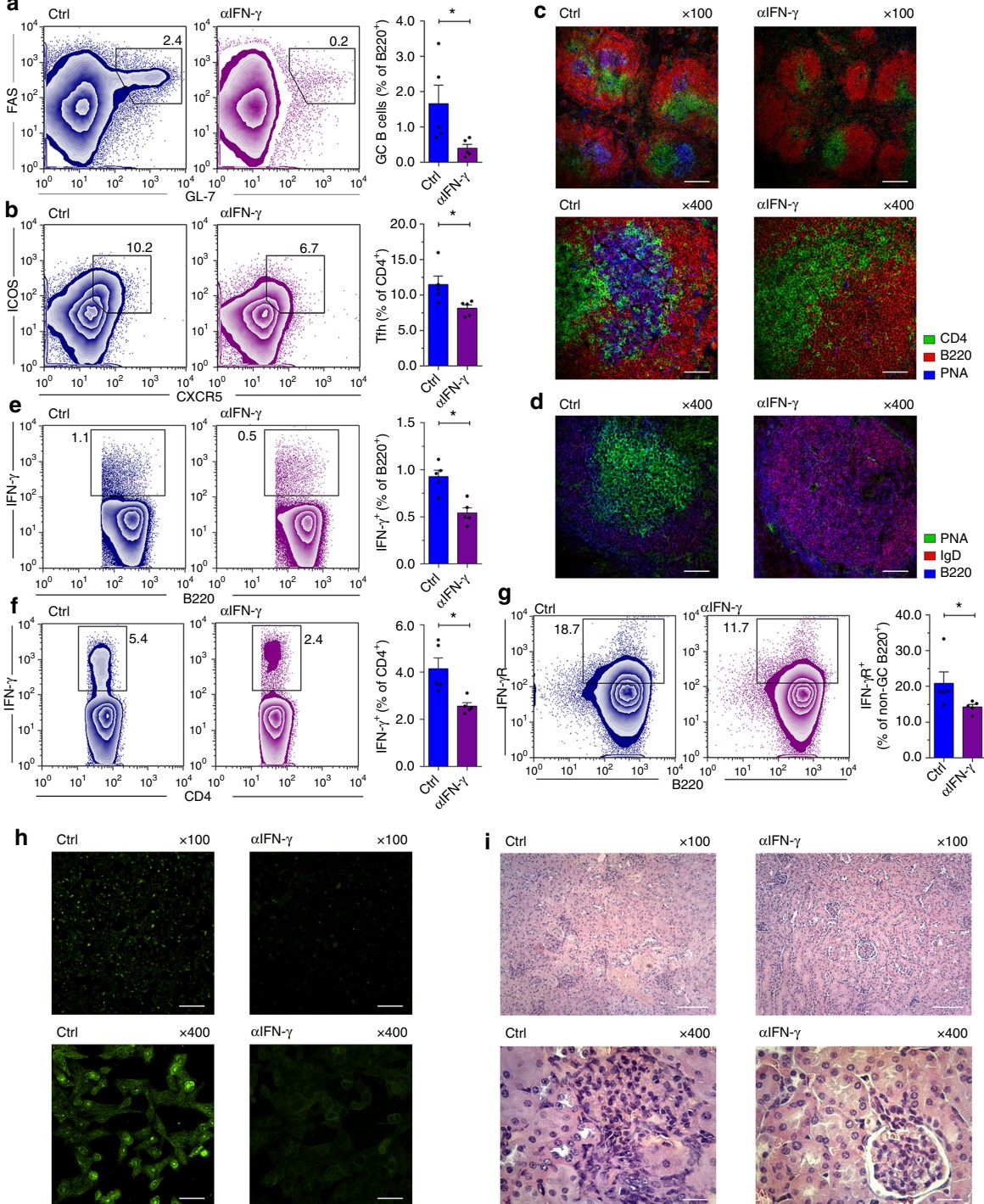

**Fig. 6** IFN-γ signaling blockade prevents spontaneous GC formation and lupus-like disease. **a**, **b** Representative plots and statistical analysis of the frequency of GC B cells and Tfh cells in the spleen of Ig-control Gal-3 KO (Ctrl) and anti-IFN-γ-treated Gal-3 KO mice (αIFN-γ), respectively (n = 5 for each group). **c** Immunofluorescence of frozen spleen sections (7 μm) from Ctrl and αIFN-γ mice stained with anti-CD4 (green), anti-B220 (red) and PNA (blue). Magnification: ×100 (top) and ×400 (bottom) (n = 5 for each group). Scale bar, 200 μm (upper panels) and 50 μm (lower panels). **d** Immunofluorescence of frozen spleen sections (7 μm) from Ctrl and αIFN-γ mice stained with PNA (green), anti-IgD (red) and anti-B220 (blue). Magnification: 4×00 (n = 5 for each group). Scale bar, 50 μm. **e** Representative plots and statistical analysis of the frequency of IFN-γ-producing B220$^+$ cells in the spleen of Ctrl and αIFN-γ mice (n = 5 for each group). **f** Representative plots and statistical analysis of the frequency of IFN-γ-producing CD4$^+$ T cells in the spleen of Ctrl and αIFN-γ mice (n = 5 for each group). **g** Representative plots and statistical analysis of the frequency of B220$^+$ GL7$^-$ FAS$^-$ non-GC IFN-γR$^+$ B cells in the spleen of Ctrl and αIFN-γ mice (n = 5 for each group). **h** ANA IgG determined with Hep-2 slides in sera from Ctrl and αIFN-γ mice. Magnification: ×100 (top) and x400 (bottom) (n = 5 for each group). Scale bar, 200 μm (upper panels) and 50 μm (lower panels). **i** Kidney sections from Ctrl and αIFN-γ mice stained with H&E. Magnification ×100 (top) and ×400 (bottom) (n = 5 for each group). Scale bar, 100 μm (upper panels) and 20 μm (lower panels). Ig-Control Gal-3 KO mice (Ctrl) are indicated in blue and αIFN-γ-treated Gal-3 KO mice (αIFN-γ) in violet. Data are the mean ± s.e.m. *p < 0.05 (unpaired Student's t-test). **a–i** Data are representative of two independent experiments

that in Gal-3 KO mice, IFN-γ overproduction leads to (induces and is required to sustain) a pathogenic and lupus-inducing GC response.

## Discussion

Our findings provide the first evidence for the involvement of Gal-3 as a key molecule in the GC reaction. Previously, we and others have found that Gal-3 regulates B cell niches in lymph node tissues and inhibits PC differentiation in vitro[17] and in vivo[45]. Our studies revealed that Gal-3 is expressed by resting B cells, as previously reported[17], and is downregulated after activation with a T-independent (LPS) or T-dependent (CD40L+IL-4) stimulus that favor B cell differentiation into PCs, suggesting that the downregulation of Gal-3 may be necessary to favor the ASC programme. Accordingly, we demonstrated that mice lacking Gal-3 have hypergammaglobulinemia and an increased frequency of ASCs in the spleen and BM. We found increased levels of IgG2c and IgG3 in Gal-3 KO mice. These Ig subclasses efficiently activate stimulatory FcγRs and complement, and they display the highest pathogenic activity of the Igs[46]. It is well known that the productions of these Ig subclasses is promoted by IFN-γ[47] and that they are closely involved in the development and progression of autoantibody-mediated autoimmune diseases, such as lupus. Here we show that, without purposeful immunization or infection, Gal-3 KO mice exhibit spontaneous GC generation in the spleen. The increase in number of GC B cells was time-dependent since GC structures appeared after 4 weeks of birth and increased with age. The increase in B cells with GC phenotype was accompanied by an expansion of Tfh cells. Additionally, the Tfh:Tfr ratio was significantly higher in these mice compared with their WT counterparts. It has been described that the balance between Treg and effector T cell is critical for proper control of the quality and magnitude of adaptive immune responses and for establishing or breaching tolerance to self- and non-self-antigens[48,49]. Therefore, the increase in the ratio of Tfh to Tfr cells in Gal-3 KO mice could have an impact on promoting the development of GCs in Gal-3 KO mice, resulting in uncontrolled GC reactions and subsequent development of an autoimmune disease[50,51].

To better understand the relationship between Gal-3 and the B cell differentiation programme, we compared gene expression levels between B cells from WT or Gal-3 KO mice. Gal-3 KO mice showed increased expression of genes related to B cell activation and GC-formation, such as Mki67, Aicda, Lilrb4, Itga1, Ig heavy chain-encoding genes and others. Interestingly, we found increased transcript levels of miRs, which coordinate GC formation, CSR/SHM, PC differentiation, and memory B cell differentiation, such as miR-155[31] and miR-217[33]. These short non-coding RNAs would individually target multiple factors in addition to regulating AID, Bcl-6 and Blimp-1. Likewise, we found quite a few miRs with as yet unidentified functions, which were differentially expressed in B cells from Gal-3 KO and WT mice and could influence GC reaction development. Given the ability of Gal-3 to bind to different molecules, such as RNA[52]; adaptor proteins involved in protein transport, such as Alix[53]; transcription factors, such as Bcl-2[54]; and components of the nuclear ribonucleoprotein complex[55], it is likely that Gal-3 could act a gene regulator, either directly or indirectly.

Analysis of molecules that have a critical role in regulating the GC reaction revealed that Gal-3 KO mice had higher percentages of CD80[+] and CD86[+] B cells, and after a TLR-stimulated B cells from Gal-3 KO mice had increased expression of CD80 in comparison to B cells from WT mice. It has been reported that B cell expression of CD80 plays a critical role in both early and late GC responses[56]. Moreover, high expression of CD80 and CD86

molecules is evident on freshly isolated B cells in patients with SLE[57]. In our model, the increased percentage of CD80 and CD86 B cells in Gal-3-deficient mice possibly facilitated Tfh induction, which in turn could allow the GC response. This hypothesis favors the induction of B cell activation and autoantibody production. It is possible that the increase in expression of CD80 and CD86 resulted from the increased concentrations of IFN-γ detected in Gal-3 KO mice rather than a direct effect of Gal-3 in regulating the expression of these proteins, as it has been reported that IFN-γ increases the expression of these molecules on B cells[58].

A question that arises is whether spontaneous GCs development was induced by the lack of Gal-3 expression specifically in B cells or if B and T cells are co-protagonists of the exacerbated GC reaction. Adoptive transfer of naive B cells into B cell-deficient µMT mice or WT mice showed that, in any case, serum levels of IgG2c and IgG3 were increased only in mice transferred with Gal-3-deficient B cells. The data suggest that B cells deficient in Gal-3 can differentiate and secrete Igs independently of a Gal-3-sufficient environment. In addition, co-cultures of naive B and T cell from WT and Gal-3 KO mice showed that stimulated Gal-3-deficient B cells produced higher levels of Igs in comparison to Gal-3-sufficient B cells and that the addition of Gal-3-deficient T cells did not increase that production. Using chimeric mice, we were able to show that the lack of Gal-3 in B cells is, by itself, sufficient to trigger disruption of immunological tolerance and to favor the generation of spontaneous GCs. By contrast, evidence provided by other groups has shown that the absence of Gal-3 would also regulate certain functions of CD4[+] T cells. Huan-Yuan Chen et al. found that Gal-3 is an inhibitory regulator of T-cell activation and functions intracellularly by promoting TCR downregulation[53]. In our work, we also observed that CD4[+] cells could produce more IFN-γ (probably by signals provided by Gal-3-deficient B cells), and we hypothesise that, although not responsible for triggering spontaneous GC formation, they contribute to maintaining and propagating the autoimmune condition mediated by IFN-γ.

Spontaneous GC B cells and Tfh cells play a crucial role in generating high-affinity pathogenic autoantibodies in many autoimmune diseases[7,10,59–62]. Accordingly, in Gal-3 KO mice, we found ANAs (including anti-dsDNA, anti-Histone and anti-SM/RNP) and kidney disease, both of which are cardinal features of systemic autoimmune diseases, such as lupus. Interestingly, Gal-3 KO mice also presented well-organised lymphoid-like structures wherein B and T cells intimately interacted within each other. This observation suggests that these T:B aggregates in the kidneys promote the local secretion of pathogenic autoantibodies that contribute to the lupus-like phenomena. Thus, Gal-3 KO mice exhibited higher concentrations of creatinine in sera and 24-h urine proteins compared with WT mice, suggesting a loss of renal function.

Interestingly, according to the serum Ig isotype profile, we observed high levels of serum IFN-γ and increased numbers of IFN-γ-producing cells in Gal-3 KO mice. Increased IFN-γ expression alone can result in the development of systemic autoimmunity, as documented by the finding that epidermal transgenic expression of IFN-γ leads to anti-dsDNA and anti-histone autoantibodies and glomerulonephritis[63]. The ability of IFN-γ to promote IgG class switching to more pathogenic autoantibodies (IgG2c and IgG3 in mice) and activation of IgG Fc receptors and complement could also contribute to disease severity[64]. Mutation of ROQUIN (Rc3h1) in Roquin^san/san mice, another model of lupus, leads to reduced decay of IFN-γ mRNA, resulting in increased IFN-γ signaling and accumulation of Tfh cells, culminating in an increase in GC B cells and autoantibodies[7]. Notably, IFN-γR plays a crucial role in the

development of autoimmune conditions, and it was clearly demonstrated recently in several mouse models[7,65,66]. Interestingly, in Gal-3 KO mice, B cells as well as T cells expressed higher levels of IFN-γR in comparison to WT, suggesting that the absence of Gal-3 conditioned both lymphocyte populations toward a better IFN-γ signaling response.

A relationship has been observed between the lack of Gal-3 and increased levels of IFN-γ. Higher serum levels of IFN-γ have been shown in melanoma-bearing Gal-3 KO mice[67], and CD4⁺ T cells incubated with Gal-3 knockdown DCs produced more IFN-γ than control cells[68]. Recently, Tseng and collaborators showed that Gal-3 may decrease IFN-γ signaling by facilitating AKT/GSK-3b/SHP2 signaling[69].

Additionally, a possible link has been proposed between commensal microbiota and autoimmunity[70]. We observed that B cells from Gal-3 KO mice were more sensitive to LPS (Supplementary Fig. 6g) than B cells from WT mice. These results, together with the observation that GCs appear in Gal-3 KO mice after the suckling period, lead us to hypothesize that the microbial flora could be a stimulus that induces B cells to differentiate into GC B cells. However, when Gal-3 KO mice were treated with a mixture of antibiotics to deplete the murine intestinal microbiota, no changes were observed in the frequency of either splenic GC B cells or Tfh in comparison to untreated mice (Supplementary Fig. 7a, b). Likewise, co-housed experiments between mutant Gal-3 KO and WT mice revealed no induction of GCs in WT mice nor prevention of spontaneous GCs in Gal-3 KO mice (Supplementary Fig. 7b), suggesting that ligands from the gut microbiota are not involved in the induction of the self-reactive GCs in Gal-3 KO mice.

Herein, we report complete dependence of the presence of self-reactive GCs on IFN-γ in Gal-3 KO mice. IFN-γ blockade limited the induction of spontaneous GC responses, reducing GC B cell and Tfh cell frequencies and Ig class switching as well as preventing the appearance of autoantibodies and kidney infiltrate; these findings demonstrated that IFN-γ overproduction was required to sustain lupus-like disease. In conclusion, we revealed a new undefined role of Gal-3 expression in controlling GC formation and, consequently, immune tolerance. Our findings demonstrate that the absence of Gal-3 in B cells intrinsically favors GC formation and highlight the potential for therapeutic targeting of this pathway in autoimmunity.

## Methods

**Mice**. All animal experiments were approved by and conducted in accordance with the guidelines of the Institutional Animal Care and Use Committee of the Facultad de Ciencias Químicas, Universidad Nacional de Córdoba (Approval Number HCD 1525/14), in strict accordance with the recommendation of the Guide to the Care and Use of Experimental Animals published by the Canadian Council on Animal Care (OLAW Assurance number A5802-01). Mice used for experiments were female and age-matched (3 and 8 weeks old and 8 months old) and housed with a 12-h light-dark cycle. C57BL/6 wild type (WT; JAX:000664) mice were obtained from The Jackson Laboratories (USA). B6.Cg-*Lgals3tm1Poi*/J (Gal-3 KO; JAX:006338), C57BL/6 CD45.1 mice (B6.SJL-*Ptprca Pepcb*/Boy; JAX:002014) and µMT mice (B6.129S2-Ighmtm1Cgn/J; JAX:002288) were obtained from The Jackson Laboratories (USA). All animals were bred and housed under barrier, temperature and humidity-controlled conditions at the Animal Facility of CIBICI-CONICET, Facultad de Ciencias Químicas, Universidad Nacional de Córdoba. Animals were euthanized with carbon dioxide ($CO_2$) followed by cervical dislocation.

**Cell culture**. Spleens were obtained and homogenised through a tissue strainer. Bone marrow (BM) cells were isolated by flushing femurs and tibias of mice with RPMI 1640. Cells were resuspended in a Tris-ammonium chloride buffer for 5 min to lyse red blood cells.

Splenic B cells were purified using a negative B cell isolation kit following manufacturers' guidelines (Miltenyi Biotec; Cat#130-049-801), or they were positively selected (B220⁺ CD3⁻ CD4⁻) by cell sorting with a FACSAria II (BD Biosciences) after staining with fluorochrome-conjugated anti-B220, anti-CD3 and anti-CD4. Both methods yielded enriched populations >93%.

For splenic CD4⁺ T cell purification, spleen cell suspensions were stained with fluorochrome-conjugated anti-CD3, anti-CD4 and anti-B220, and CD3⁺ CD4⁺ B220⁻ T cells were sorted with a FACSAria II (BD Biosciences). For co-culture experiments, spleen cell suspensions were stained with fluorochrome-conjugated anti-CD4, anti-B220 and anti-GL7, and CD4⁺ B220⁻ GL7⁻ T cells and B220⁺ CD4⁻ GL7⁻ B cells were obtained by sorting with a FACSAria II (BD Bioscience) and were >95% pure.

B1 cells were purified from total peritoneal cells obtained from washes of the peritoneal cavity of WT mice with 2% FBS-PBS and stained with fluorochrome-conjugated anti-mouse CD19, anti-mouse B220, anti-mouse CD3 and anti-mouse CD11b antibodies. B1 cells were positively selected by CD19⁺ B220⁺ CD11b⁺ CD3⁻ expression with a FACSAria cell sorter (BD Biosciences).

Follicular B2 cells (B220⁺ CD21⁺ CD23^high CD3⁻CD4⁻) were obtained from spleen of WT mice by cell sorting using fluorochrome-conjugated anti-mouse B220, anti-mouse CD21, anti-mouse CD23 antibodies, anti-mouse CD3, and anti-mouse CD4.

The specific cell population and gating strategy in each case are described in Supplementary Fig. 8.

All cell cultures were performed in complete RPMI: RPMI 1640 (Gibco, Life Technologies) medium supplemented with 2 mM glutamine (Gibco, Life Technologies), 50 µM 2-ME (Sigma), and 40 µg/ml gentamicin (Fabra Laboratories) containing 10% FBS (PAA).

In some experiments, B cells ($2 \times 10^6$) were cultured in complete RPMI in the absence or presence of LPS (Invivogen, 5 µg/mL) for different times (12, 24, 48 and 72 h) or with LPS or anti-CD40 (3 µg/ml) plus rmIL-4 (10 ng/ml) for 72 h. CD4⁺ T cells were cultured in complete RPMI at 37 °C in a 96-well plate (Corning) precoated with anti-CD3 (Biolegend, 1 µg/mL) plus anti-CD28 (Biolegend, 2 µg/mL) for 72 h.

**Flow cytometry**. For surface staining, single-cell suspensions were washed in ice-cold FACS buffer (PBS-2% FBS) and incubated with fluorochrome labelled-antibodies for 20 min at 4 °C. For intracellular cytokine staining, cells were stimulated for 5 h with 50 ng/mL PMA (phorbol 12-myristate 13-acetate), 500 ng/mL ionomycin and GolgiStop (BD Biosciences). After surface staining, cells were fixed and permeabilized with Cytofix/Cytoperm according to the manufacturer's instructions (BD Biosciences; Cat#554714). For intracellular Foxp3, Bcl-6, Blimp-1, and Ki-67 staining, a Foxp3 Fix/Perm kit was used (eBiosciences; Cat#00-5523-00) after surface staining. Data were collected on a FACSCanto (BD Biosciences) and were analysed with FlowJo software (TreeStar).

**Antibodies**. The following anti-mouse antibodies were used for FACS (Supplementary Table 1): B220-FITC (RA3-6B2), B220-PE (RA3-6B2), B220-PE-Cy7 (RA3-6B2), CD3-PerCP-Cy5.5 (145-2C11), CD86-APC (GL-1), CD4-APC-Cy7 (GK1.5), CD23-FITC (B3B4), IFN-γ-FITC (XMG1.2), Bcl-6-PE (7D1), IL-21R-APC (4A9), Streptavidin-PerCP-Cy5.5, NK1.1-PE-Cy7 (PK136), TCR-β-chain-APC (H57-597), ICOS-PE-Cy7 (C398.4A), and IFN-γRα-Biotin (2E2) were from Biolegend; B220-APC (RA3-6B2), CD11b-FITC (M1/70), CD21-PE (eBio8D9), PD-1-PE (J43), CD4-PE (GK1.5), CXCR4-PE (2B11), CD40-PE (1C10), CD69-PerCP-Cy5.5 (H1.2F3), Foxp3-PerCP-Cy5.5 (FJK-16s), Foxp3-APC (FJK-16s), IFN-γ-APC (XMG1.2), Ki-67-eFlur660 (SolA15), GL7-eFluor660 (GL7), CD11c PE-Cy7 (N418), MHC Class II I-Ab PerCP-eFluor710 (AF6-120.1), CD49b PE (DX5), CTLA-4 PE (UC10-4B9) and CD39 PerCP-eFluor710 (24DMS1) were from eBioscience; CD11c-FITC (HL-3), CD45.1-APC-Cy7 (A20), CD45.2-PerCP-Cy5.5 (104), IL-17-PE (TC11-18H10), Bcl-6-PE-Cy7 (K112/91), CD80-PE (16-10A1), CD138-APC (281-2), FAS (CD95)-PE (Jo2), FAS (CD95)-PE-Cy7 (Jo-2), FAS-Biotin (Jo2), Streptavidin-PE, CXCR5-Biotin (2G8) and IgD FITC (11-26c.2a) and CD38 PE (90) were from BD Biosciences; and Blimp-1-PE (C-21) was from Santa Cruz.

The following anti-mouse antibodies were used for tissue immunofluorescence: PNA-FITC was from Vector Laboratories; PNA-Alexa Fluor 647, CD4-Alexa Fluor 488 (RM4-5), CD3-Alexa Fluor 647 (500A2) and IgG(H + L)-Alexa Fluor 594 were from Invitrogen; IgD-Alexa Fluor 594 (11-26c.2a) was from Biolegend, B220-PE (RA3-6B2) and B220-APC (RA3-6B2) were from eBioscience.

**In vitro co-culture of T and B cells**. Sorted $1.10^5$ naïve T cells (CD4⁺ GL7⁻ B220⁻) from WT or Gal-3 KO mice were co-cultured with $1.10^5$ naïve B cells (B220⁺ GL7⁻ CD4⁻) from WT or Gal-3 KO mice in a 96-well plate in the presence of 2 µg/mL soluble anti-CD3 (2C11) and 5 µg/mL anti-IgM (Jackson Immunoresearch). After 6 days of culture, the Igs concentration in the supernatants was measured by ELISA.

**Western blot analysis**. To detect Gal-3, 35 µg whole cell lysates in radio-immunoprecipitation assay buffer were mixed with 6 µL sample buffer and boiled for 5 min. Cell lysates were resolved by 12.5% SDS-PAGE. Monoclonal anti-Gal-3 B2C10 (diluted 1:500) (Santa Cruz Biotechnology) was used to detect Gal-3. As a control for protein loading, nitrocellulose membranes were reproved with 0.25 mg/mL anti-β-actin antiserum (Santa Cruz Biotechnology) or with a polyclonal anti-calreticulin antibody (Santa Cruz Biotechnology) (diluted 1:2000). The levels of

proteins were analysed by Scion Image 1.62C alias. Uncropped original scans of immunoblots are provided in Supplementary Fig. 9.

**Adoptive transfer.** Sorted B cells from CD45.2 WT or CD45.2 Gal-3 KO mice were transferred intravenously into recipient mice. B cells ($2 \times 10^7$/mouse) were transferred when μMT mice were the recipients and $6 \times 10^6$ B cells were transferred per mouse when CD45.1+ mice were the recipients. Mice were rested for 2 weeks and bled to determine serum IgG2c and IgG3 concentrations.

**Generation of BM chimeras.** For the BM chimeras shown in Fig. 3d, C57BL/6 recipient mice were lethally irradiated with 100 Gy ($^{137}$Cs source) and reconstituted via intravenous injection with a combination of an 80:20 mixture of $10.10^6$ BM cells from B6.129S2-Ighm$^{tm1Cgn}$/J (μMT) plus C57BL/6 (WT) or (μMT) plus Gal-3 KO donors, respectively. For the BM chimeras shown in Fig. 3h, CD45.2 C57BL/6 recipient mice were lethally irradiated with 100 Gy and reconstituted via intravenous injection with a combination of a 30:70 mixture of $10 \times 10^6$ BM cells from B6.Cg-Lgals3$^{tm1Poi}$/J (Gal-3 KO) and CD45.1 C57BL/6 (CD45.1- Gal-3$^{+/+}$) donors, respectively. For both chimeras, experiments were performed 8–10 weeks after reconstitution.

**Anti-IFN-γ treatment.** Three-week-old female Gal-3 KO mice were bled prior to treatment and then injected ip with 500 μg anti-IFN-γ monoclonal (BioXcell, clone: XMG 1.2) (anti-IFN-γ treated Gal-3 KO mice) or 500 μg rat IgG1 Isotype control mAb (BioXcell) (Ig-control Gal-3KO mice) every 3 days for 3 weeks.

**Antibiotic treatment.** Three-week old mice were treated for 5 weeks with van-comycin (500 mg/L; Cat#V1130-1G), neomycin sulphate (1 g/L; Cat#N6386-25G), and metronidazole (1 g/L; Cat#M3761-25G) in drinking water (all from Sigma-Aldrich). Antibiotic-containing water was changed twice a week.

**Immunoglobulin and cytokine quantification.** Total IgM and IgG isotype levels were determined by ELISA in sera. Plates were coated with 10 μg/mL of the isotype-specific goat anti-mouse antibody (IgM, IgG1, IgG2c, IgG2b, and IgG3; SouthernBiotech) overnight at 4 °C and blocked with 1% BSA. Sera were incubated overnight at 4 °C. Peroxidase-conjugated anti-mouse IgM or IgG isotypes were added and incubated for 1 h at 37 °C. The reaction was developed with TMB Substrate Reagent (BD Biosciences), followed by quantification on a Microplate Reader 450 from Bio-Rad. The concentration was measured with reference to standard curves using known amounts of the respective murine immunoglobulin isotypes (SouthernBiotech).

Cytokines were determined by ELISA in mouse serum and in the supernatant of sorted splenic T cells (CD4$^+$ B220$^-$) from WT and Gal-3 KO mice cultured in vitro with anti-CD3 plus anti-CD28 for 72 h. ELISA was performed with paired antibodies from eBioscience for mouse IFN-γ (Cat#88-7314-77), IL-21 (Cat#88-8210-88), IL-4 (Cat#88-7044-77), IL-6 (Cat#88-7064-88) and IL-2 (Cat#88-7024-77), according to the manufacturer's instructions.

**ELISPOT assay.** For analysis of the production of ASC by ELISPOT, plates were coated with 10 μg per well of goat anti-mouse Ig (H + L) (Southern Biotech) in sodium bicarbonate buffer and then blocked with 2% BSA/PBS. Cells from spleens or from BM were serially diluted across the plate and then incubated for 4–6 h at 37 °C. Biotin-goat anti-Ig, goat anti-IgM, or goat anti-IgG (Southern Biotech) diluted in blocking buffer was added, followed by three washes with 0.1% Tween 20 and a secondary incubation with ExtrAvidin-alkaline phosphatase (Sigma-Aldrich). Spots were detected using BCIP/NBT (Sigma-Aldrich) and scanned and counted with an ImmunoSpot Analyzer (Cellular Technology).

**Immunofluorescence analysis of B cell suspensions and tissues.** Purified B cells cultured in medium or with LPS (10 μg/mL) for 72 h were washed once with PBS and spun down onto a slide using a Cyto-Spin system. For the fixation procedure, B cells were treated with 3% *p*-formaldehyde in PBS for 20 min at RT. Endogenous fluorescence was quenched by incubation with 10 mM NH₄Cl. B cells were permeabilized with 0.5% Nonidet P-40 diluted in PBS pH 7.4 for 20 min at RT. Nonspecific binding was blocked, and then B cells were incubated with FITC-labelled anti-Gal-3 diluted in DAKO Diluent with Background Reducing Components solution (Dako, Carpinteira, CA, USA). DAPI was used for nuclear counterstaining. Cells were mounted with DAKO Fluorescent Mounting Medium (Dako), and immunofluorescence was evaluated using a Nikon Eclipse TE-2000U microscope (Nikon Corporation; Tokyo, Japan).

For tissue immunofluorescence, spleens from WT and Gal-3 KO mice were collected and frozen over liquid nitrogen. Frozen sections with a thickness of 7 μm were cut, fixed for 10 min in cold acetone, left to dry at 25 °C and stored at −80 °C until use. Slides were hydrated in TRIS buffer and blocked for 30 min at 25 °C with 10% normal mouse serum in TRIS buffer. After blockade, slides were incubated for 50 min at 25 °C with the different antibodies diluted in TRIS Buffer. Slices were mounted with FluorSave (Merck Millipore). Images were collected with an Olympus microscope (FV1000), and those recorded in the far-red channel were pseudo-coloured blue.

**ANA assessment.** For ANA detection, serum was serially diluted in PBS from 1:20 to 1:1280 for indirect immunofluorescence on fixed HEp-2 slides (BioSystems S.A.) and for indirect immunofluorescence of fixed *Crithidia luciliae* slides (Antibodies, Inc.) serum diluted 1:20 for anti-dsDNA antibody detection. Slides were incubated with 20 μL of serum for 45 min, followed by an incubation with Alexa Fluor 488-labelled goat anti-mouse IgG (Invitrogen) for 20 min. Slides were observed with an Olympus microscope (FV1000) at different magnifications.

**Measurements of specific autoantibodies.** For specific auto-Ab, 96-well plates were precoated overnight at 4 °C with 100 μg/mL calf thymus dsDNA (D3664-5 × 2MG; Sigma-Aldrich), 5 μg/mL Sm/RNP (ATR01-10; Arotec Diagnostic Limited) or 5 μg/mL histone (ATH01-02; Arotec Diagnostic Limited). Plates were blocked for 1 h with 1% BSA in PBS before addition of diluted serum for 2 h. Specific antibodies were detected using goat anti–mouse IgM-HRP, IgG-HRP, IgG1-HRP, IgG2c-HRP, or IgG3-HRP (SouthernBiotech), and peroxidase reactions were developed using TMB Substrate Reagent (BD Biosciences), followed by quantification at 450 nm on a Microplate Reader 450 from Bio-Rad. The results are expressed in optical density (OD) units.

**Histology.** Kidneys were fixed in 10% neutral-buffered formalin and embedded in paraffin. Seven-micron-thick paraffin-embedded sections were dewaxed and stained with haematoxylin and eosin (H&E). The images were captured using a digital camera Carl Zeiss Microimaging GmbH, with Moticam 2000. Each kidney was examined at 400× magnification and scored from 0 to 12 based on the following features: glomerular size and hypercellularity (0–4), changes in glomerular matrix (0–4) and the degree of hypercellularity in the tubulointerstitium (0–4). Pathology was scored by observers who were blinded to the genotype.

For kidney IF staining, kidneys from WT and Gal-3 KO mice were collected and frozen over liquid nitrogen. Frozen sections with a thickness of 7 μm were cut, fixed for 10 min in cold acetone, left to dry at 25 °C and stored at −80 °C until use. Slides were hydrated in TRIS buffer and blocked for 30 min at 25 °C with 10% normal mouse serum in TRIS buffer. After blockade, slides were stained with anti-B220-Alexa Fluor 488, anti-CD45-Alexa Fluor 555 and anti-CD4-Alexa Fluor 647 in TRIS Buffer. Slices were mounted with FluorSave (Merck Millipore). Images were collected with an Olympus microscope (FV1000).

**Urea and creatinine determination.** Urea and creatinine concentrations were determined on a Dimension RxL Max system (enzymatic/colorimetric method and Jaffé reaction, respectively; Siemens Healthcare Diagnostics), following the manufacturer's instructions.

**Proteinuria.** Urine samples were collected 24 h by placing mice in custom-made mouse metabolic cages. Urine total protein was determined on a Dimension RxL Max system (colorimetric method; Siemens Healthcare Diagnostics), following manufacturer's instructions.

**DNA microarrays.** For gene-expression analysis, purified B cells from 8-week-old WT and Gal-3 KO mice were lysed with TRIzol reagent. Total RNA was extracted with the RNAeasy Mini Kit (Qiagen). RNA quality was verified in an Agilent Bioanalyser and measured with a Nanodrop 1000 (Thermo Scientific). Microarray experiments were performed on GeneChip Mouse Gene 2.1 ST arrays (Affymetrix). The working lists were created by filtering probes with detection P values <0.05 for all chips and discarding overlapping probes. Each dataset was derived from two biologically independent replicate samples. Independent samples were compared by computing fold ratios and filtered at a 1.2-fold threshold for Venn diagrams and pathway analysis. For pathway analysis, GenBank accession numbers were mapped to the Ingenuity database (IPA, http://www.ingenuity.com) to retrieve relevant biological processes. Heat maps were generated using the web interface programme Matrix2png.

**Statistical analysis.** All data are expressed as mean ± s.e.m. Replicates were biological replicates and *n* represents the number of individual mice analysed per experiment. Mice were assigned to groups at random for all mouse studies and, where possible, mixed among cages. Careful consideration has been given to calculate the number of mice needed to give statistically significant results, while using the minimum number of mice possible. Numbers of mice for each experiment were decided after consultation with a statistician at UNC. Experiments were performed independently in triplicate to control for experimental variation. No exclusion was performed. Statistical comparisons were determined using unpaired or paired two-tailed Student's *t*-test or one-way analysis of variance (ANOVA) followed by Tukey–Kramer multiple comparison test when appropriate. Statistical analyses were performed using GraphPad Prism software (GraphPad). Data followed Gaussian distribution with similar variance between groups that were compared. Differences were considered statistically significant when the P values <0.05.

**Data availability.** The microarray data have been deposited in the ArrayExpress database under accession number E-MTAB-6053. The authors declare that the data supporting the findings of this study are available within the article and

its Supplementary Information files, or are available upon reasonable requests to
the authors.

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

## Acknowledgements

We thank N.G. Núñez, M.P. Abadie, M.P. Crespo, V. Blanco, F. Navarro, D. Lutti, C. Mas, M.C. Sampedro and L. Almada for their excellent technical assistance. This work was supported by grants from the Agencia Nacional de Promoción Científica y Técnica (PICT 2013–2007), CONICET and the Secretaría de Ciencia y Técnica-Universidad Nacional de Córdoba. E.V.A.R., C.L.M., J.M., O.C. and A.G. are researchers from CONICET. C.G.B., F.F.V., M.C.R., J.T.B., M.G.S. thank CONICET for the fellowship awarded.

## Author contributions

C.G.B. performed and designed most of the experiments, analysed the data and prepared the figures and manuscript. M.C.A.V. contributed experimentally to the data presented in Figs. 1 and 2, and R.C.G. performed the experiments shown in Fig. 1 and Supplementary Fig. 1. F.F.V., M.C.R., J.T.B. and M.G.S. helped with the experiments. J.M. and O.C. helped with the chimera mouse experiments. P.E. contributed to the Affymetrix microarrays. C.L.M. and E.V.A.R. contributed to the study design and analysis and corrected the manuscript. A.G. conceived, designed and supervised the study and wrote the manuscript. All of the authors reviewed the manuscript before submission.

## Additional information

**Competing interests:** The authors declare no competing interests.

