## [Peer Review File · Nature Communications]

Reviewers' comments:

Reviewer #1 (galectin-3, signalling)(Remarks to the Author):

The authors found Gal-3 KO mice exhibited a high frequency of GC B cells and T follicular helper cells. This phenotype was correlated with an increased percentage of Ab-secreting cells and high concentrations of IgM, IgG2c, IgG3, and IFN- γ in serum. Using an adoptive transfer approach, the authors were able to contribute the phenotypes to galectin-3 deficiency in B cells. Moreover, they noted Gal-3 KO mice developed autoimmunity responses, such as mononuclear cell infiltrations in the kidney and antinuclear Ab secretion. They showed that the development of these phenotypes could be reversed by IFN- γ blockade in Gal-3 KO.

Major comments:

1. A major issue is whether the authors used littermate controls in their experiments. This issue was highlighted in a review by Holmdahl and Malissen "The need for littermate controls" in *Eur. J. Immunol.* 2012. 42: 45–47.

2. It is known that mice spontaneously develop autoantibodies, as indicated by the measurement of anti-nuclear antibodies. For example, Nusser et al. showed that most mice of the C57BL/6 (B6) strain spontaneously produced IgG ANA at 8–12 months of age. There, the ANA titers of over 1:1000 were noted (with a mean titer of 453.9) (*Eur. J. Immunol.* 2014. 44: 2893–2902). In this manuscript, the authors only used sera diluted at 1:40. At this dilution, one is likely to see nonspecific staining, and thus false positive results. The authors should do serial dilutions and also use a larger number of mice.

3. In the studies by Nusser et al., the authors also found spontaneous occurrence of IgM deposition in kidneys and lymphocyte infiltrates in submandibular salivary glands in 8-12 month old C57BL/6 mice. Thus, for the studies shown in Fig. 5 and 6, the authors need to show more quantitative measurements and use a larger number of mice.

4. For the finding of kidney damage in Gal-3 KO mice, because Gal3 is expressed in many cells and tissues, the data presented do not allow the conclusion that the phenotype is contributed by lack of galectin-3 in B cells. To reach a definitive conclusion, the authors need to study mice following B cell transfer, as they employed to look at IgG, IgM, and IFN-g production.

Other comments:

1. The authors found LPS dramatically reduced Gal3 expression in B cells. However, the authors' own group previously reported sgalectin-3 was not expressed in B cells from normal mice, but is induced in B cells activated by LPS, IL-4 and CD40 cross-linking. The authors need to explain the discrepancy.

2. To the authors' credit, they use Gal-3 KO mice and cells to reach the conclusion on the functions of endogenous galectin-3, unlike a large number of papers that use exogenously added galectin-3 to derive at the conclusions, which might not represent those of endogenous galectin-3. In the latter case, the authors would inevitably conclude that galectin-3 functions extracellularly. In the current manuscript, the functions the authors determined for endogenous galectin-3 could be due to the protein functioning intracellularly (although with the data they presented, they are not able to definitely prove this). It would be useful to discuss this issue briefly.

Reviewer #2 (SLE, nephritis, plasma/B cells)(Remarks to the Author):

Galectin family members have been implicated in autoimmunity and deficiency of galectin 3 has been shown both to increase and decrease disease severity in a number of models of autoimmune disease. Previously described mechanisms include inhibitory effects in the immune synapse resulting in disturbances in T cell cytokine production with altered ratio of IFN gamma to IL10, effects on Tregs and effects on T cell apoptosis. B cell effects include altered B cell development, increased plasma cell differentiation and altered regulation of anergic B cells. Galectin 3 also is involved in regulation of glucose homeostasis and may play a role in tissue repair and remodeling. Both pathogenic and protective roles for galectin 3 have been reported in different diseases.

Galectin 3 knockout mice have previously been reported to develop obesity and systemic inflammation as well as impaired hematopoiesis and impaired glucose tolerance. Relevant to lupus, a defect in macrophage phagocytosis of apoptotic cells has been reported. Interestingly, galectin 3 deficient kidneys have been reported to be protected from tubular damage and interstitial fibrosis in renal transplant models and similarly galectin 3 deficiency protects from lung fibrosis and NASH in animal models. Nevertheless aged galectin 3 deficient mice develop proteinuria with glomerulosclerosis and interstitial damage.

In this manuscript the authors more closely examine the effects of galectin 3 in B cells. The same galectin 3 mice reported in previous studies were used. They show that galectin 3 is clearly downregulated in activated B cells and its expression is substantially decreased in germinal center and plasma cells compared with naïve B cells. It appears to restrain the differentiation of these cells as Gal3 KO mice have an increase in GC cells and PC after immunization. Indeed Gal3 KO mice have spontaneous GCs and increased numbers of TFH and develop antinuclear antibodies as they age. Using adoptive transfer experiments they show that this is a B cell intrinsic feature of Gal3 KO B cells. Nevertheless there also appears to be a T cell intrinsic effect as shown in Figure 3 and Figure 4. As previously reported the T cells make high levels of IFN gamma. Thus, both B cell intrinsic and extrinsic factors are involved in the enhanced GC formation in the Gal KO mice. They go on to show that this is dependent on IFN gamma. As previously reported by Rahman and colleagues, IFN gamma appears to play a B cell intrinsic role in GC development. They do not address the question of loss of tolerance in pre-GC B cell compartments and this should be acknowledged in the discussion.

The authors then suggest that the Gal KO mice develop a lupus like disease. However they do not show much data to support this. The pathology is not well described and is different to that previously reported in these mice (glomerulosclerosis and interstitial tubular disease). There needs to be quantitation both of the morphologic changes and of the cellular infiltrates. Are there lupus related autoantibodies? Is there antibody deposition in the glomeruli? Or complement? Is there proteinuria? Is the lifespan of the mice affected? Are other organs affected? Not surprisingly, the systemic inflammation they describe depends on IFN gamma.

Overall, apart from the inadequate studies of the lupus phenotype the manuscript is well done and the B cell intrinsic effects of Gal3 described here are novel and convincing. The role of Gal3 in T cells is underplayed in the discussion and should be addressed. Therapeutic targeting of this molecule will clearly be fraught with dangers given the pleiotropic effects of Gal3 and this should be discussed in the last paragraph.

Minor points

1. Figure 2B, given the small error bars are there no statistically significant differences from 3 or more experiments with $n > 5$ mice? Please check? It will be more informative to the readers if the authors replace bar graphs with scatter bar graphs
2. The right hand graph of Figure 3K does not look significant whereas the first bar graph of Figure S5D looks significant. Could the authors check?

3. Figure 4D, do the Tfh cells also have the highest IFN γ expression?
4. WT cells should be shown as controls for Figure 4G and H and I.
5. Figure S6C, right flow plots, please check labeling. Should be CD86?

C I B I C I

Córdoba, Argentina, February 11th, 2018

Associate Editor

Dear Editor , thanks for your email dated January 31, 2018 explaining that you are ready to proceed to the editorial decision about our manuscript NCOMMS-17-15669 entitled "Galectin-3 deficiency drives lupus-like disease by promoting spontaneous germinal centers formation via IFN γ " but for that we need to edit the manuscript following Nature Communications instructions and your suggestions. Accordingly,

- 1) We indicated the text changes of the second version of the manuscript (answering reviewer's comments) in red, while the changes of the last revision of the manuscript (third version) were marked using the track change feature in Word.
- 2) All figure legends were included in the text of the manuscript.
- 3) We followed the Nature Communications style guide: Abstract limit (150 words), subheading length in Results as well as Methods (60 characters including space) etc. In our last version of the manuscript the main text length was reduced from 6200 to 5500 words.
- 4) The abstract was adhered to Nature style with two sentences for background, followed by "Here we report that...". We stated the results and conclusions, and lastly we introduced one sentences of a broad-implication summary.
- 5) The name of the statistical tests used were stated in the Method section and also in the legend for each figure, including supplemental figure legends.
- 6) We include a new Supplementary Figure (Supplementary Figure 7) that was originally present only in the point-by-point response but not in the original main manuscript. This figure is now cited properly in the main text of the last version.
- 7) All 'data not shown' of our original version of the manuscript were deleted, because that information was not essential to explain the role of Galectin-3 in spontaneous germinal center formation and lupus-like disease development.
- 8) Finally, according to the observation of Reviewer#01 we deleted the sentences explaining results obtained with the injection of recombinant Galectin-3 in Gal-3KO mice and these information was not included a new data.

C I B I C I

Now, our manuscript comply with the format requirements detailed in the Nature Communication checklist for authors at:

http://www.nature.com/article-assets/npg/ncomms/authors/ncomms_manuscript_checklist.pdf

We appreciated all suggestions that improved the last version of our manuscript and we would be glad to respond to any further questions and comments that you may have,

Looking forward hearing from you soon,

Sincerely,

Adriana Gruppi, PhD

Centro de Investigaciones en Bioquímica Clínica e Inmunología.
Facultad de Ciencias Químicas. Universidad Nacional de Córdoba.

REVIEWERS' COMMENTS:

Reviewer #1 (Remarks to the Author):

Discussion:

The authors stated, “---we asked ourselves if the treatment with recombinant exogenous Gal-3 could reverse the evidenced phenomenon. We observed that after treatment with recombinant Gal-3, mice were not able to normalize IFN γ levels, nor repress induction of Tfh, nor reduce the frequency of GC B cells (although a slight but not significant reduction is observed) (Fig. S7A-C). These experiments highlight once again the differing functions of exogenous Gal-3 and their endogenous counterparts”.

The administration of recombinant Gal-3 into experimental mice really does not allow any conclusions to be made with regard to the mode of action of the protein (intracellular vs extracellular), because one is not able to predict where in treated mice the injected protein will reach and act. This reviewer suggests that the authors delete these sentences and not include the new data.

Reviewer #2 (Remarks to the Author):

The authors have addressed all the major points raised by the reviewers and I have no more comments

C I B I C I

Córdoba, Argentina, March 28th, 2018

To whom it may concern

Dear Referees,

Thank you very much for a thorough evaluation on our manuscript NCOMMS-17-15669A entitled "Galectin-3 deficiency drives lupus-like disease by promoting spontaneous germinal centers formation via IFN- γ ". We have agreed with almost all your comments and we have revised our manuscript accordingly. We have included all your suggestions and clarifying the manuscript when needed. Thank you for your suggestions that allowed us to greatly improve the quality of our work. Your comments are in bold text and our responses in plain italics.

Reviewer #1 (Remarks to the Author):

The authors stated, “we asked ourselves if the treatment with recombinant exogenous Gal-3 could reverse the evidenced phenomenon. We observed that after treatment with recombinant Gal-3, mice were not able to normalize IFN γ levels, nor repress induction of Tfh, nor reduce the frequency of GC B cells (although a slight but not significant reduction is observed) (Fig. S7A-C). These experiments highlight once again the differing functions of exogenous Gal-3 and their endogenous counterparts”. The administration of recombinant Gal-3 into experimental mice really does not allow any conclusions to be made with regard to the mode of action of the protein (intracellular vs extracellular), because one is not able to predict where in treated mice the injected protein will reach and act. This reviewer suggests that the authors delete these sentences and not include the new data.

We considered that the reviewer's suggestions are valid, so we have decided not to incorporate the new data regarding the administration of recombinant Gal-3 in Gal-3 KO mice.

Reviewer #2 (Remarks to the Author):

The authors have addressed all the major points raised by the reviewers and I have no more comments

Thank you for your comments.

C I B I C I

We would be glad to respond to any further questions and comments that you may have,

Looking forward hearing from you soon,

Sincerely,

Adriana Gruppi

Centro de Investigaciones en Bioquímica Clínica e Inmunología.
Facultad de Ciencias Químicas. Universidad Nacional de Córdoba.